# How Does the Health Literacy of Adults Residing in Social Housing Compare with That of Those Living in Other Housing Tenures in Australia? A Secondary Analysis of the Australian National Health Survey 2017–2018 Dataset

**DOI:** 10.3390/ijerph20186753

**Published:** 2023-09-13

**Authors:** Megan Freund, Natasha Noble, Allison Boyes, Matthew Clapham, David Adamson, Robert Sanson-Fisher

**Affiliations:** 1Health Behaviour Research Collaborative, College of Health, Medicine and Wellbeing, University of Newcastle, Callaghan, NSW 2308, Australia; natasha.noble@newcastle.edu.au (N.N.); allison.boyes@newcastle.edu.au (A.B.); rob.sanson-fisher@newcastle.edu.au (R.S.-F.); 2Equity in Health and Wellbeing Research Program, Hunter Medical Research Institute, Newcastle, NSW 2305, Australia; 3Clinical Research Design and Statistics Support Unit, Hunter Medical Research Institute, Newcastle, NSW 2305, Australia; matthew.clapham@hmri.org.au; 4Home in Place Co Ltd., Hamilton, NSW 2303, Australia; davida@homeinplace.org; 5Community and Social Policy, Faculty of Business and Creative Industries, University of South Wales, Treforest Campus, Pontypridd CF37 1DL, UK

**Keywords:** health literacy, disadvantage, social housing, housing tenure, socioeconomic disparities in health, public housing

## Abstract

Background: Social housing tenants have poorer health outcomes than homeowners or those renting privately. Health literacy is associated with access to care and health outcomes. This study aimed to examine the health literacy of Australian adults residing in social housing compared with that of people living in other housing types. Methods: A secondary analysis of the Australian National Health Survey 2017–2018 dataset was undertaken. A total of 5275 respondents were included in the sample and completed the Health Literacy Questionnaire (HLQ). Respondents were categorised according to their housing tenure: 163 (3.1%) respondents were living in social housing, 873 (17%) were living in private rentals, 2085 (40%) were homeowners, and 2154 (41%) were homeowners/mortgages. Mean scores were calculated for each of the nine health literacy domains in the HLQ and compared across housing tenure using linear regression models. Results: Social housing tenants had lower mean domain scores than either homeowners, owner mortgagees, or private renters on six of the nine health literacy domains. This included ‘having sufficient information to manage my health’, ‘social support for health’, ‘ability to engage with healthcare providers’, ‘navigating the healthcare system’ ‘ability to find good health information’, and being able to ‘understand health information enough to know what to do’. However, the differences in mean scores were small. Conclusions: Increasing health literacy may be an important part of multicomponent interventions seeking to improve the health and wellbeing of social housing tenants.

## 1. Introduction

Health literacy concerns the capacity of people meet the complex demands of health in a modern society [1]. It encompasses an individual’s ability to seek, locate, comprehend and appraise health information, and apply the knowledge gained to address or solve a health problem [2]. An individual’s health literacy skills are crucial for making health-related decisions [3]. People with low health literacy are more likely to have poorer health outcomes such as low engagement with health services (including preventive healthcare), poorer medication adherence, and greater engagement in health risk behaviours [4,5,6].

There is a well-established relationship between social disadvantage and poorer health outcomes—with factors such as access to education, adequate housing, employment, income, and being of an Indigenous, linguistically, or culturally diverse background being associated with health status [7]. A growing body of research suggests that health literacy is an independent predictor of health outcomes, even after controlling for other social determinants of health [8,9,10]. Low health literacy can exacerbate underlying health access and equity issues for marginalised or vulnerable groups [11].

People residing in social housing represent a particularly marginalized group. Social housing is secure and affordable rental housing provided by not-for-profit, non-government or government organisations to assist people who are unable to access suitable accommodation in the private rental market. Around 4.2% (790,000 people) of the Australian population reside in social housing. In Australia, social housing represents a scarce resource and as such is being increasingly allocated to people with complex needs [12,13]. This includes those with physical and psychosocial disability, at risk of homelessness, or on very low incomes [14]. In 2020–2021, 81–86% of the Australian social housing (public and community housing) allocation was to tenants in ‘greatest need’ [15]. Social housing tenants are consistently found to have poorer health compared with those living in other housing types [16,17], as well as higher levels of health risk behaviours such as smoking [18].

To date, very limited published research has investigated the health literacy of social housing residents. Agarwal et al. 2018 examined health literacy in a sample of older adults (n = 237) living in subsidized housing buildings in Canada. Functional health literacy was assessed using the Newest Vital Sign UK (NVS-UK), where scores can range from 0 to 6. Scores of 0–3 represent inadequate health literacy [19]. They found that over 82% of participants had inadequate health literacy levels [19]. Lower health literacy levels were associated with higher body mass index, less than a college or university education (compared with high school or less) and having pain and/or discomfort [19]. In 2006 in Australia, the Adult Literacy and Life Skills Survey (ALLS) reported that almost 60% of adults aged between 17 and 74 years did not have adequate health literacy skills to effectively and efficiently understand and apply health-related information in their daily lives [20]. The ALLS assessed functional aspects of health literacy, such as understanding text, finding information in documents, and problem-solving capabilities [21]. Health literacy in Australia is now assessed using the Health Literacy Questionnaire, which moves beyond a functional approach and considers a broader range of health literacy characteristics [21]. A cross-sectional study of people attending health and community care organisations conducted in Victoria, Australia, reported lower scores on some health literacy domains associated with not speaking English at home, being born overseas, lower education levels, and not having private health insurance [22]. However, to our knowledge, no previous work has specifically explored how the health literacy of social housing tenants differs from those in other housing tenures. A recent scoping review of Australian health literacy studies noted recent efforts to focus on exploring health literacy among Aboriginal communities but did not identify any studies within the social housing setting [11].

Given the potential impact of poor health literacy on the health outcomes of a significant proportion of the population who reside in social housing, more research is needed to help understand the health literacy strengths and challenges of this marginalised group. This evidence will be useful to policy and decision makers and social housing providers when considering initiatives to improve the health of social housing residents, who represent one of the most marginalised and disadvantaged sectors the community. Therefore, this study aimed to examine, using the 2017–2018 Australian National Health survey data, the health literacy of Australian adults (≥18 years of age) residing in social housing compared with that of adults living in other housing types (owner/owner mortgage/private rental).

## 2. Methods

### 2.1. Study Design

This study was based on a secondary analysis of the National Health Survey (NHS) 2017–2018 dataset. The NHS is an Australia-wide survey conducted every two to five years by the Australian Bureau of Statistics (ABS) [23]. The survey is designed to collect a range of information about the health and wellbeing of Australians, with the 2017–2018 NHS being the most recent survey with data available for analysis [23]. The NHS uses a stratified multistage approach to select a representative random sample of private dwellings to complete the survey [23]. Further details regarding the survey methodology are available in the NHS Users’ Guide for 2017–2018 [24]. Relevant data were securely provided for secondary analysis through the ABS Data Laboratory (DataLab).

### 2.2. Sample and Procedure

The 2017–2018 NHS included a sample of approximately of 21,315 persons from 16,384 private dwellings across Australia [23]. Non-private dwellings (e.g., hotels, hospitals, nursing homes, short-stay caravan parks) were excluded from the survey [23]. Trained ABS interviewers conducted personal interviews with selected residents in sampled dwellings. One adult (aged 18 years and over) in each dwelling was selected and interviewed about their own health characteristics and provided information about the household (e.g., household income, composition) [23]. The selected adult respondent also provided health information about one child in the household (or some children aged 15–17 years were interviewed with parental consent). The health literacy survey was conducted on a sub sample of 5790 respondents aged over 18 years who had participated in the NHS. Only data for adult respondents were considered in the secondary analysis. Data for 16,376 households were included in National Health Survey data. Of these, 7224 were asked to complete the health literacy questions. Health literacy scores could not be calculated for 1434 participants due to participants not fully responding or selecting not applicable. A valid housing status was not available for a further 515 participants.

### 2.3. Outcome Measures

**Housing type.** Survey respondents completed a question about housing tenure (e.g., whether the dwelling is owned/rented). Social housing residents are identifiable in the dataset through the landlord type (state or territory housing authority).

**Health literacy.** The Health Literacy Questionnaire (HLQ) collects information on how people find, understand, and use health information and how they manage their health and interact with healthcare providers. Developed and extensively tested in Australia, the HLQ consists of 44 questions assessing nine domains of health literacy [25]. Respondents indicate their level of agreement with a set of health literacy statements (‘strongly agree’, ‘agree’, ‘disagree’, or ‘strongly disagree’) or the perceived difficulty of a health literacy characteristic (‘always easy’, ‘usually easy’, ‘sometimes difficult’, ‘usually difficult’, or ‘cannot do or always difficult’). Scores range between 1 to 4 (for first 5 domains) and 1 to 5 (for domains 6 to 9). A domain score is calculated by summing the scores within each domain and dividing this value by the number of items in the respective domain. Higher scores indicate higher health literacy. As all nine domains comprising the HLQ are considered independent, and it is not recommended to calculate an overall HLQ score [25]. The domains of the HLQ comprise: feeling understood and supported by healthcare providers; having sufficient information to manage my health; actively managing my health; social support for health; appraisal of health information; ability to actively engage with heath care providers; navigating the healthcare system; ability to find good health information; and understand health information well enough to know what to do.

**Demographic characteristics.** The 2017–2018 NHS collected data on a range of demographic characteristics including age, gender, postcode, employment status, education level, main language spoken at home, and household composition (lives with adults and children; lives with children only; lives with other adults only; lives alone).

## 3. Analysis

All statistical analyses were completed in SAS v9.4 (SAS Institute, Cary, NC, USA). Mean scores were calculated for each of the nine health literacy domains in the HLQ. For domains 1–5, the mean scores ranged from 1 to 4 (1 = strongly disagree; 4 = strongly agree). For domains 6–9, the mean scores ranged from 1 to 5 (1 = cannot do or always difficult; 5 = always easy). Population weighted linear regression models were used to compare mean scores on each domain across housing types, controlling for age, sex, socioeconomic disadvantage, and remoteness area. 

Participant postcodes were used to determine socioeconomic status and remoteness. Socioeconomic status was determined according to the Socio-Economic Indexes for Areas (SEIFA) index. SEFIA scores are standardised against a mean of 1000 with a standard deviation of 100 [26]. Remoteness areas were classified using the Accessibility and Remoteness Index of Australia (ARIA+), as either major city, inner regional, outer regional, or remote [27]. Age and SEIFA scores were modelled using natural cubic splines with knots at percentiles 5, 27.5, 50, 72.5, and 95%. Housing types were categorised as: social housing (dwelling rented from a state or territory housing authority); private rental (dwelling rented from a real estate agent); homeowner (dwelling owned by someone in the household); and homeowner/mortgagee (dwelling owned by someone in the household with a mortgage or loan). Other housing types (such as dwelling rented from a parent or relative not in the same household or from the owner or manager of a caravan park, etc.) were excluded from analysis. Replicate weights were used so standard errors and 95% confidence intervals account for the sampling design. Person weights were used, and estimates are relative to the total adult population residing in the four housing types in 2017–2018, approximately 17 million.

## 4. Results

### 4.1. Sample Description

After the removal of participants who did not complete the HLQ and who did not have a valid housing status, a final sample of 5275 participants was available for analysis. Of these, 163 (3.1%) participants were in social housing, 873 (17%) were in private rentals, 2085 (40%) were homeowners, and 2154 (41%) were homeowners/mortgages. The population-weighted characteristics of the sample by housing type are provided in Table 1.

### 4.2. Comparison of Health Literacy by Housing Type

Table 2 presents means and standard errors for the weighted health literacy domain scores by housing type. The domain scores ranged from 2.8 to 4.0 for social housing, 2.9 to 4.2 for private rental, and 2.9 to 4.3 for both owner mortgagee and owners.

Table 3 presents the health literacy domain regression results. Higher health literacy domain scores were indicated for residents of all other housing types compared with social housing tenants for four of the health literacy domains. This included ‘social support for health’, ‘ability to engage with healthcare providers’, ‘ability to find good health information’, and being able to ‘understand health information enough to know what to do’. The mean difference ranged from 0.13 to 0.35. For two domains (‘navigating the health system’ and ‘having sufficient information to manage my health’), only owner mortgagees had higher health literacy scores compared with social housing tenants. The mean differences were 0.18 and 0.14, respectively.

## 5. Discussion

This study found that social housing tenants had lower mean domain scores than adults residing in other housing types for six of nine health literacy domains. The differences in mean domain scores on the HLQ were relatively small, indicating only minor differences in health literacy between housing groups. Social housing tenants had statistically significantly lower mean scores for ‘having sufficient information to manage my health’ (where low scores on this construct indicate feeling that there are many gaps in their knowledge and that they do not have the information they need to live with and manage their health concerns [25]), ‘social support for health’ (where low scores indicate feeling alone and unsupported for health [25]), ‘ability to actively engage with healthcare providers’ (where low scores indicate a passive approach to healthcare, and inability to ask questions to get information or to clarify what they do not understand [25]), ‘navigating the healthcare system’ (where low scores indicate an inability to advocate on their own behalf and a limited understanding of services or supports available [25]), ‘ability to find good health information’ (where low scores indicate dependence on others to offer information [25], and ‘understand health information well enough to know what to do’ (where low scores reflect difficulties with being able to understand written information, including numerical information, in relation to health [25]).

We identified only one previous study that examined the health literacy of people living in social housing (Agarwal l et al. 2018) [19]. However, unlike the current study, Agarwal et al. 2018 did not compare the health literacy of social housing residents to that of residents of other housing types. The lower health literacy scores reported in the current study are broadly consistent with results reported by Agarwal et al. 2018. They reported that more than 80% of 237 older adult residents (>55 years of age) living in subsidised housing had below adequate health literacy levels using the Newest Vital Sign UK—validated version to measure health literacy [19]. However, these findings cannot be compared directly with those of the current study, given the differences in methodologies, including the tool used to measure health literacy and the age of the targeted sample. Furthermore, the HLQ used to measure health literacy in the current study does not provide guidance as to whether health literacy domain scores indicate adequate or inadequate levels of health literacy. Rather, the HLQ aims to provide a profile of health literacy competencies or needs [25]. The current study findings indicate a range of potential health literacy needs among social housing tenants as compared with residents in other housing types.

The finding that social housing residents had lower health literacy domain scores is compatible with previous research that examined health literacy by indicators of disadvantage. A 2015 Australian study examined the factors associated with health literacy of clients attending health and community care organisations using the HLQ [22]. Similar to the present results, the study did not find a consistent difference in health literacy across all HLQ domains according to demographic characteristics of the sample, but did report significant differences in some mean domain scores according to sex, age, education level, having private health insurance, being born in Australia or overseas, speaking English at home, and living along. The largest differences in health literacy domain scores were seen for English not being spoken at home and being born overseas. Like the current study, smaller differences in mean domain scores were found for other indicators of socioeconomic disadvantage including lower levels of education and not having private health insurance.

There is consistent research evidence that as a group, people who are from disadvantaged backgrounds tend to have lower health literacy compared to those who are less disadvantaged [28]. Although the current study did find lower health literacy scores for some domains, the differences were relatively small. One explanation may be that a relatively high proportion of the social housing tenants included in the study had an education level of completed high school or higher (46%). Health literacy and literacy are related concepts and strong literacy skills help to understand and use health information and services [7]. Another explanation may relate to the utility of the HQL measure of health literacy across population subgroups that are inclusive of marginalised groups [28]. Some authors have questioned whether the available measures of health literacy are able to detect true differences in capacities and skills in marginalized groups [28]. In addition, in the current study health literacy scores could not be calculated for a relatively large proportion of participants asked to complete the health literacy questions due to missing data (approximately 20%), and a valid housing status was not available for a smaller proportion of participants (approximately 7%). It is possible that these participants may have been more disadvantaged, living in social housing, and had lower health literacy, but were not reflected in the final sample available for analysis.

## 6. Implications for Future Research and Practice

Further research is required to more fully understand the small differences in health literacy found in this study. This could include more research examining the health literacy of social housing tenants to see if the health literacy differences in the current study can be replicated elsewhere. Such research should also examine the effects of health literacy on access to healthcare and health outcomes. Future research should also examine the reliability and validity of existing measures of health literacy measures for use with marginalised groups such as social housing tenants [28]. Despite the small differences found, future initiatives that aim to improve the health and wellbeing of social housing tenants should include strategies to improve health literacy, with a focus on social support for health, engaging with healthcare providers, and support for finding good health information and understanding health information. This should be part of a multicomponent initiative that also addresses the complex, multiple, and interrelated issues faced by tenants of social housing that also impact health (e.g., employment and training opportunities and social isolation).

The findings of the current study should be considered in light of its limitations and strengths. Participants who did not select a valid housing status or who indicated that the health literacy questions were not applicable were not included in the analysis, potentially affecting representativeness of the housing type samples. The research was reliant on measuring health literacy via the HLQ which, although validated in many participant groups, has not been validated in such marginalised groups as social housing residents. A significant strength of the study is the nationally representative sample, although the sample size of social housing tents was relatively small.

## 7. Conclusions

The current study found that people residing in social housing have lower health literacy on a subset of domains compared with adults residing in other housing types. The relatively small difference in health literacy found between social housing tenants and those in other housing types was unexpected. Given this, more research is required to understand the health literacy strengths and weaknesses of social housing tenants and the impacts on access to healthcare and health outcomes for this population group. As part of multicomponent initiatives, increasing health literacy will be an important part of improving the health and wellbeing for this group.

## Figures and Tables

**Table 1 ijerph-20-06753-t001:** Weighted sociodemographic characteristics.

Category	Social Housing (N = 163) Weighted % (SE)	Private Rental (N = 873) Weighted % (SE)	Owner (N = 2154) Weighted % (SE)	Owner Mortgagee (N = 2085) Weighted % (SE)
**Age**				
18–24	11% (4.74)	17% (1.58)	7% (1.01)	12% (0.99)
25–34	4.2% (1.93)	34% (1.95)	8% (1.02)	20% (0.88)
35–44	13% (3.94)	24% (1.50)	5% (0.77)	25% (0.89)
45–54	14% (3.79)	14% (1.29)	10% (0.77)	24% (0.81)
55–64	25% (4.90)	6% (0.76)	22% (0.89)	14% (0.66)
65–74	21% (5.27)	3.8% (0.66)	26% (0.79)	4.3% (0.44)
75+	11% (2.95)	1.4% (0.39)	21% (0.53)	0.7% (0.21)
**Gender**				
Female	56% (6.45)	49% (2.06)	54% (1.16)	49% (1.05)
**Employment**				
Employed	14% (5.12)	75% (2.23)	41% (1.45)	84% (1.21)
Unemployed	8% (2.74)	3.2% (0.82)	0.8% (0.24)	3.3% (0.53)
Not in labour force	79% (5.11)	22% (2.17)	58% (1.46)	13% (1.13)
**Education**				
University	6% (2.32)	32% (2.00)	22% (1.47)	36% (1.35)
Diploma/certificate	24% (4.63)	31% (2.01)	29% (1.39)	33% (1.21)
Year 12	16% (4.83)	15% (1.79)	12% (1.21)	16% (1.05)
Year 10 or 11	19% (4.16)	12% (1.38)	19% (0.89)	9% (0.73)
Lower/not determined	35% (5.65)	10% (1.22)	17% (0.98)	7% (0.68)
**SEIFA**				
1–2 (lowest)	37% (5.19)	20% (2.38)	17% (1.29)	12% (1.19)
3–4	21% (5.15)	21% (2.20)	21% (1.79)	19% (1.79)
5–6	18% (5.16)	19% (2.32)	19% (1.39)	21% (1.52)
7–8	18% (4.44)	21% (1.99)	21% (1.62)	24% (1.47)
9–10 (highest)	6% (2.39)	20% (1.92)	22% (1.69)	23% (1.36)
**ARIA+**				
Major cities	71% (4.84)	76% (1.63)	68% (1.60)	76% (1.00)
Inner regional	13% (3.47)	18% (1.75)	23% (1.59)	17% (1.02)
Outer regional/remote	15% (3.30)	6% (0.96)	9% (0.72)	7% (0.51)
**Mainly speak English at home**				
Yes	88% (4.89)	80% (2.00)	94% (1.18)	88% (1.20)
**Housing composition**				
Lives with adults and children	13% (5.24)	36% (2.51)	15% (1.44)	56% (1.77)
Lives with children only	10% (3.13)	6% (0.78)	1.1% (0.32)	2.8% (0.39)
Lives with other adults only	15% (4.21)	34% (2.65)	46% (1.84)	22% (1.32)
Could not be determined	11% (3.88)	11% (1.40)	20% (1.58)	13% (1.26)
Lives alone	51% (5.02)	14% (1.28)	18% (1.07)	7% (0.50)

SE = standard error; SEIFA = Socio-Economic Indexes for Areas; ARIA+ = Accessibility and Remoteness Index of Australia.

**Table 2 ijerph-20-06753-t002:** Weighted health literacy scores by domain and housing type.

Health Literacy Domain	Social Housing (N = 163) Weighted Mean (SE) *	Private Rental (N = 873) Weighted Mean (SE) *	Owner Mortgagee (N = 2154) Weighted Mean (SE) *	Owner (N = 2085) Weighted Mean (SE) *
Feeling understood and supported by healthcare providers	3.1 (0.05)	3.1 (0.02)	3.2 (0.02)	3.2 (0.01)
Having sufficient information to manage my health	3.0 (0.06)	3.2 (0.02)	3.2 (0.01)	3.2 (0.01)
Actively managing my health	3.0 (0.05)	3.1 (0.02)	3.1 (0.01)	3.1 (0.01)
Social support for health	3.0 (0.05)	3.2 (0.02)	3.2 (0.01)	3.2 (0.01)
Appraisal of health information	2.8 (0.07)	2.9 (0.02)	2.9 (0.01)	2.9 (0.02)
Ability to actively engage with heath care providers	3.9 (0.09)	4.1 (0.03)	4.2 (0.02)	4.2 (0.02)
Navigating the healthcare system	3.9 (0.08)	4.0 (0.03)	4.0 (0.02)	4.1 (0.02)
Ability to find good health information	3.8 (0.09)	4.1 (0.03)	4.1 (0.02)	4.1 (0.02)
Understand health information well enough to know what to do	4.0 (0.08)	4.2 (0.03)	4.3 (0.02)	4.3 (0.02)

* Scores range between 1 and 4 (for first 5 scales) and 1 and 5 (for scales 6 to 9); SE = standard error.

**Table 3 ijerph-20-06753-t003:** Comparison of health literacy domain scores by housing type.

Domain	Category (vs. Social Housing)	Weighted Estimate (95% CI) *	*p*
Feeling understood and supported by healthcare providers	Owner	0.082 (−0.017 to 0.18)	0.101
	Owner mortgagee	0.088 (−0.002 to 0.178)	0.054
	Private rental	0.003 (−0.091 to 0.097)	0.955
Having sufficient information to manage my health	Owner	0.115 (−0.002 to 0.231)	0.053
	Owner mortgagee	0.143 (0.018 to 0.268)	**0.026**
	Private rental	0.097 (−0.031 to 0.225)	0.135
Actively managing my health	Owner	0.101 (−0.004 to 0.205)	0.059
	Owner mortgagee	0.069 (−0.028 to 0.165)	0.159
	Private rental	0.054 (−0.057 to 0.165)	0.334
Social support for health	Owner	0.209 (0.108 to 0.311)	**<0.001**
	Owner mortgagee	0.214 (0.116 to 0.313)	**<0.001**
	Private rental	0.134 (0.039 to 0.23)	**0.007**
Appraisal of health information	Owner	0.059 (−0.068 to 0.186)	0.353
	Owner mortgagee	0.071 (−0.065 to 0.208)	0.301
	Private rental	0.057 (−0.083 to 0.197)	0.421
Ability to actively engage with heath care providers	Owner	0.296 (0.122 to 0.469)	**0.001**
	Owner mortgagee	0.345 (0.152 to 0.537)	**<0.001**
	Private rental	0.261 (0.073 to 0.449)	**0.007**
Navigating the healthcare system	Owner	0.163 (−0.003 to 0.329)	0.054
	Owner mortgagee	0.183 (0.004 to 0.362)	**0.045**
	Private rental	0.162 (−0.007 to 0.331)	0.060
Ability to find good health information	Owner	0.273 (0.089 to 0.457)	**0.004**
	Owner mortgagee	0.283 (0.089 to 0.477)	**0.005**
	Private rental	0.236 (0.049 to 0.423)	**0.014**
Understand health information well enough to know what to do	Owner	0.27 (0.094 to 0.446)	**0.003**
	Owner mortgagee	0.268 (0.101 to 0.434)	**0.002**
	Private rental	0.212 (0.056 to 0.368)	**0.009**

* All estimates adjusted for age, sex, SEIFA, and ARIA; Significant differences are shown in bold.

## Data Availability

Some summary statistics from the National Health Survey (NHS) 2017–18 dataset are publicly available. However, those seeking access to the data would need to seek approval for virtual access to the detailed microdata files that remain in the secure Australian Bureau of Statistics environment.

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
