# Peer review of "How Does the Health Literacy of Adults Residing in Social Housing Compare with That of Those Living in Other Housing Tenures in Australia? A Secondary Analysis of the Australian National Health Survey 2017–2018 Dataset"

_ijerph, 2023, doi:10.3390/ijerph20186753_

Round 1
Author Response
Please see the attached response to Reviewer 1.

Reviewer 2 Report
Thank you for asking me to review this paper which reports the findings from a secondary analysis of the National Health Survey dataset examining health literacy in adults residing in social housing. Although the determinants of health literacy have been well documented research investigating specific factors remains limited. Your paper will contribute to the emerging body of evidence.
I have the following comments and questions:
Introduction:
Overall, the manuscript lack some clarity and a structure that tells ‘the story’. It is at times difficult to follow and weakens your paper and the point you intend to make. There are sections in which the authors seem to contradict previous statements, some references could be preplaced with more recent work and the original reference is always preferred over a secondary reference. See more details below.
For example:
The author’s reference for HL was published in 2005. There are more current and comprehensive definitions available. See Sorensen’s work.
Line 62-68
People residing in social housing represent a particularly marginalized group. Social housing is secure and affordable rental housing provided by not-for-profit, non-government or government organisations to assist people who are unable to access suitable accommodation in the private rental market. Around 4.2% (790,000 people) of the Australian population reside in social housing. In Australia, social housing represents less than 4% of housing supply and is inevitably, as a scarce resource, is being increasingly allocated to people with complex needs.
Reviewer’s comment:
This section is confusing. It sounds as if 4% of housing supply is social housing and that 4.2% of people in Australia live in social housing. Does it mean that 0.2% do not have access. I would have assumed this number to be higher and hence, I wonder whether my interpretation is not correct, hence please review.
Line 75-76
Agarwal et al 2018 examined health literacy in a sample of older adults (n = 237) living in subsidized housing buildings in Canada. They found over 82% of participants had below adequate health literacy levels.
Reviewer’s comment:
Please provide more detail such as how they measured HL and how they reached this result.
Line 79-81
In Australia, it is estimated that almost 60% of adults aged between 17 and 74 years do not have adequate health literacy skills to effectively and efficiently understand and apply health-related information in their daily lives
Reviewer’s comment:
Where possible you should cite the original source, which in this case was the ALLS (2006) reported by the ABS
Discussion:
The discussion fails to critically engage with the findings of the study.
There are number of reasons that would explain the results in the discussion.
E.g. how accurate is SEIFA as a determinant of socio-economic position? Please describe why is is used but also it shortcomings
The differences in use of HL measures. THE HLQ is a self-reported assessment whereas the Newest Vital Sign ‘tests’ individuals understanding of a health-related text. There is some debate about the measures and how accurately they measure HL and a critical engagement is necessary to analyse the findings.
A further question needs to be considered for the Australian context. If in 2006 about 60% of the adult population reported low HL, why is the data from 2017-2018 so different? There are many reasons as to why the results seem to be so different to those of other studies and these need to be discussed.
Line 201- 207
This study is one of very few that has examined the health literacy of adults residing in social housing. We found that social housing tenants had lower levels of health literacy for six of nine domains. The difference in means domain scores were relatively small (<0.5) indicating only minor difference between housing groups in health literacy.
As this, to our knowledge, is the first study to compare health literacy levels of social housing tenants to those living in other housing types, we cannot compare our findings to similar previous research. The lower levels of health literacy reported in the current study is consistent with previous research examining health literacy in social housing.
Reviewer’s comment:
These statements contract each other.
There are a few wordings/expression that could be chaned.
line 44 the word "factor' is used to describe difficulties following medical recommendation - A more fitting prescription may be 'ability' or capacity or skills.
Line 52 - ... and Indigenous and cultural background. Is this section referring to individuals from Indigenous and linguistically and culturally diverse backgrounds or Indigenous cultural heritage?
Line 55 - ... greater chances of worsening health. Better descriptor might be 'risk'
Author Response
Please see the attached response to Reviewer 2.

Round 2
Reviewer 1 Report
Dear Authors,
Thank you for answering my questions. I accept all of your answers and additions. I have only one suggestion about the Introduction part:
"4000 words or more to meet the suggested Journal word length. As a result, the authors have not shortened the Introduction section as part of this revision. However, the authors are willing to do so, if the Editor also recommends this change."
I think it would be better if you lengthen the manuscript in the Results and Discussion part and not in the Introduction. This part of the manuscript is board in this way. Please reconsider this part of the manuscript.
Best regards,
